# KLF4 Regulates Metabolic Homeostasis in Response to Stress

**DOI:** 10.3390/cells10040830

**Published:** 2021-04-07

**Authors:** Andrew Blum, Kate Mostow, Kailey Jackett, Estelle Kelty, Tenzing Dakpa, Carly Ryan, Engda Hagos

**Affiliations:** Department of Biology, Colgate University, Hamilton, NY 13346, USA; ablum@colgate.edu (A.B.); kmostow@colgate.edu (K.M.); kjackett@colgate.edu (K.J.); ekelty@colgate.edu (E.K.); tdakpa@colgate.edu (T.D.); caryan@colgate.edu (C.R.)

**Keywords:** KLF4, metabolism, Warburg effect, autophagy, GLUT1

## Abstract

Cancerous cells are detrimental to the human body and can be incredibly resilient against treatments because of the complexities of molecular carcinogenic pathways. In particular, cancer cells are able to sustain increased growth under metabolic stress due to phenomena like the Warburg effect. Krüppel-like factor 4 (KLF4), a context-dependent transcription factor that can act as both a tumor suppressor and an oncogene, is involved in many molecular pathways that respond to low glucose and increased reactive oxygen species (ROS), raising the question of its role in metabolic stress as a result of increased proliferation of tumor cells. In this study, metabolic assays were performed, showing enhanced efficiency of energy production in cells expressing KLF4. Western blotting showed that KLF4 increases the expression of essential glycolytic proteins. Furthermore, we used immunostaining to show that KLF4 increases the localization of glucose transporter 1 (GLUT1) to the cellular membrane. 2′,7′-Dichlorodihydrofluorescein diacetate (H_2_DCF-DA) was used to analyze the production of ROS, and we found that KLF4 reduces stress-induced ROS within cells. Finally, we demonstrated increased autophagic death in KLF4-expressing cells in response to glucose starvation. Collectively, these results relate KLF4 to non-Warburg metabolic behaviors that support its role as a tumor suppressor and could make KLF4 a target for new cancer treatments.

## 1. Introduction

Krüppel-like factor 4 (KLF4) is a zinc-finger-containing transcription factor involved in the regulation of cellular growth, proliferation, differentiation, invasion, autophagy, and embryogenesis [1,2,3,4,5]. In addition to its diverse regulatory functions, KLF4 was discovered to be one of four factors required for reprogramming differentiated fibroblasts into induced pluripotent stem cells [6]. In clinical studies, KLF4 is frequently found to be either overexpressed or underexpressed in cancer cells, both of which are correlated with poor prognosis [7,8]. KLF4′s effect on cancer has been found to be context dependent due to differential regulation between cell types as well as its ability to act as both a transcriptional activator and a repressor [9,10]. However, in many human cancers, KLF4 is regarded as a tumor suppressor. For instance, the expression of KLF4 is downregulated in bladder, lung, pancreatic, colorectal, gastric, esophageal, and prostate cancers [11,12,13,14,15]. In vitro, mouse embryonic fibroblasts (MEFs) lacking KLF4 are more proliferative, in addition to being more prone to chromosome aberrations and centrosome amplification [16]. This genomic instability can be corrected by reintroduction of KLF4 into MEFs lacking KLF4 [17]. Recently, KLF4 has been found to regulate mitophagy and mitochondria dynamics, indicating a relationship between the transcription factor and cellular metabolism [18,19].

Cancer cells tend to have a metabolism characteristic of the Warburg effect, where the cells are increasingly reliant on glycolysis, producing anti-apoptotic by-products [20,21]. It is important to note that the Warburg effect is an oversimplification of cancer metabolism and that metabolic dysregulation can be multifaceted in cancer cells [22]. Altered metabolism in cancer cells, especially in cells that are increasingly reliant on glycolysis, is known to support a more evasive response to chemotherapy [23]. Understanding how KLF4 controls cellular metabolism may give rise to new therapeutic treatments that cannot be evaded.

Some research has been performed on KLF4′s regulation of metabolism, but little is known about this process and how it may relate to cancer. KLF4 overexpression in iPSC reprogramming promoted glycolysis, while decreasing the tricarboxylic acid cycle through activation of TCL1 [24]. In breast cancer, where KLF4 has been characterized as an oncogene, KLF4 directly induced transcription of phosphofructokinase, enhanced glucose intake, and increased lactate production [25].

Altered glucose intake is a common metabolic impairment in cancerous tissue and is often achieved by dysregulation of glucose transporter 1 (GLUT1) [26]. GLUT1 is one of the most commonly expressed membrane proteins and is critical for passive glucose uptake [27]. GLUT1 is held in intracellular stores and is quickly released during metabolic stress to facilitate the rapid intake of glucose [26]. GLUT1 trafficking to the outer membrane has been visualized in live cancer cells in response to glucose starvation and may serve as a proxy for the efficiency of GLUT1-dependent glucose intake [28].

Increased reliance of cancer cells on glycolysis makes glucose starvation a particularly interesting cancer metabolism experiment. Cancer cells typically experience increased cell death as compared to normal cells when exposed to low glucose conditions [29,30]. However, tumors can quickly develop resistance to starvation due to the selective pressures of increased cell death [31]. Since KLF4 is a known regulator of cell death, it could serve as an important regulatory gene in response to glucose starvation [10].

To develop a deeper understanding of KLF4′s role in the metabolism of cancer cells, we aimed to characterize how KLF4 responds to metabolic stressors, and the molecular pathways underlying these responses. We first used metabolic assays to reveal differences in macroscale metabolic behavior and found that KLF4 increases glycolytic capacity as well as shifts cells to a less Warburg-like metabolism. Using Western blots, qPCR, and immunostaining, we characterized the potential mechanisms through which KLF4 regulates metabolism and cell death in response to glucose starvation. We found that KLF4 upregulates key glycolytic proteins as well as facilitates GLUT1 outer membrane localization. Furthermore, since KLF4 has been previously shown to regulate reactive oxygen species (ROS), we analyzed whether KLF4 responds to glucose starvation due to an increase in ROS [18,32]. Finally, we found that KLF4 induces autophagic cell death in response to glucose starvation. These findings allow for a greater understanding of KLF4′s role as a context-dependent tumor suppressor and may provide insight into potential cancer therapies that target KLF4′s widespread regulation of cell metabolism.

## 2. Materials and Methods

### 2.1. Cell Culture, Reagents, and Drug Treatments

Mice heterozygous for *Klf4* alleles (*Klf4*+/−) on a C57BL/6 background were crossbred [33]. Mouse embryonic fibroblasts (MEFs) that are wild type (*Klf4*+/+), heterozygous (*Klf4*+/−), or null (*Klf4*−/−) for *Klf4* were derived from day 13.5 embryos. The RKO (RKO-EcR-KLF4) cell line was derived from a human colon cancer cell line and stably transfected with the *pAdLoxEGI-KLF4* plasmid, as previously described [34]. KLF4 is conditionally expressed in RKO cells via the addition of 5 µM Ponasterone-A (PA) in DMSO for 3 days. Control cells were treated with the equivalent amount of DMSO. Both MEFs and RKO cells were grown in Dulbecco’s Modified Eagle’s Medium (DMEM) with 10% fetal bovine serum (FBS) and 1% penicillin/streptomycin at 37 °C in an atmosphere containing 5% CO_2_. Cells were passed every 3 days at a density of 10^6^ cells. Cell morphology was assessed by a microscope in *Klf4*+/+ and *Klf4*−/− MEFs treated with either full media or no glucose.

### 2.2. Analysis of Metabolic Parameters

Wild-type and *Klf4-*null MEFs as well as DMSO-treated and PA-treated RKO cells were seeded at 20,000 cells/well in XF24 cell culture plates. After a 24 h incubation period, the growth medium was removed from each well. Cells were washed twice. Cells were then incubated at 37 °C without CO_2_ for 1 h. ATP levels and metabolic parameters were measured using the Seahorse XF96 Analyzer (Agilent, Santa Clara, CA, USA). The oxygen consumption rate (OCR) and the extracellular acidification rate (ECAR) were detected under basal conditions followed by the sequential addition of glucose, oligomycin, and 2-deoxy-D-glucose (2DG) (*N* = 3).

### 2.3. Western Blot Analysis

Cell protein extraction and Western blot analyses were performed, as previously described [4]. Nitrocellulose membranes were immunoblotted with the following primary antibodies: β-actin, KLF4 (Cell Signaling, Danvers, MA, USA), hexokinase2 (HK2), monocarboxylate transporter 4 (MCT4), pyruvate kinase M2 (PKM2), GLUT1 (Abcam, Cambridge, UK), microtubule-associated proteins 1A/1B light chain 3B (LC3), and caspase-3 (Santa Cruz Biotechnology, Dallas, TX, USA). The blots were then incubated with appropriate horseradish-peroxidase-conjugated secondary antibodies (Cell Signaling, Danvers, MA, USA) for 1 h at room temperature. The antibody–antigen complex was visualized using ImageLab (Bio-Rad, Hercules, CA, USA). ImageLab was also used to quantify the Western band intensity (*N* = 4).

### 2.4. Immunostaining

GLUT1 antibodies were used to identify membrane localization. DMSO-treated RKO cells, PA-treated RKO cells, and wild-type and *Klf4-*null MEFs were grown to 70% confluency on glass slides. One sample of each cell type was incubated in DMEM containing 0% glucose (no glucose (NG)) for 12 h, while another sample of each was incubated in normal DMEM (full media (FM)) for 12 h. Cells were fixed in 3.7% formaldehyde in DMEM for 15 min. Cells were then blocked in a solution of 3% bovine serum albumin and 0.1% TRITON X100 for 1 h. GLUT1 antibody was used followed by secondary anti-rabbit Alexa Fluor-555 antibody in RKO cells and anti-rabbit Alexa Fluor-488 antibody in MEFs. RKO cells and MEFs were counterstained with DAPI. Cells were then imaged with a Zeiss 710 confocal laser scanning microscope. GLUT1 was scored blinded by rating the degree of membrane localization relative to cytosolic GLUT1 expression as absent, slight, moderate, or heavy. The slight and moderate categorizations were combined into one category called some localization. Data were quantified using Pearson’s chi-square test (*N* = 3), with 200 MEF cells and 400 RKO cells counted.

### 2.5. qPCR

Total RNA from cultured *Klf4*+/+ and *Klf4*−/− MEFs was isolated using the RNeasy1 Mini Kit (Qiagen, Valencia, CA, USA) according to the manufacturer’s protocol. RNA was subjected to gDNA elimination columns (Qiagen, Valencia, CA, USA) in order to remove any contaminating genomic DNA. cDNA was prepared from 500 ng of RNA and amplified with the Omniscript RT Kit (Qiagen) and polyT primer (Integrated DNA Technologies, Coralville, IA, USA). The synthesized cDNA was subjected to RT-qPCR analysis using the SYBR^®^ Green PCR Master Mix for 40 cycles (Life Technologies) following the manufacturer’s protocol. The expression of PKM2, MCT4, or GLUT1 was normalized to the expression level of β-actin. The relative fold change in the gene expression level was calculated by comparing the normalized gene expression in *Klf4*−/− MEFs to that in *Klf4*+/+ MEFs or comparing the RKO cells treated with either DMSO or PA. The data shown represent two independent experiments, each performed in triplicate. PCR reactions were performed using the following primers purchased from Integrated DNA Technologies (Coralville, IA, USA): F: 5′ ATG GAG GGG AAT ACA GCC C and R: 5′ TTC TTT GAA GCT CCT TCG TT for ACTIN; F: 5′ TCG CAT GCA GCA CCT GAT and R: 5′ CCT CGA ATA GCT GCA AGT GGT A for mouse PMK2 [35]; F: 5′ ATG GCT GAC ACA TTC CTG GAG C and R: 5′ CCT TCA ACG TCT CCA CTG ATC G for human PMK2; F: 5′ TGT TAG TCG GAG CCT TCA TT and R: 5′ CAC TGG TCG TTG CAC TGA ATA for mouse MCT4 [36]; F: 5′ CGT TCT GGG ATG GGA CTG AC and R: 5′ ATG TGC CTC TGG ACC ATG TG for human MCT4 [37]; F: 5′ TCA ACA CGG CCT TCA CTG and R: 5′ CAC GAT GCT CAG ATA GGA CAT C for mouse GLUT1 [38]; and F: 5′ CTG CTC ATC AAC CGC AAC and R: 5′ CTT CTT CTC CCG CAT CAT CT for human GLUT1 [39]. RT-PCR fold changes were calculated using 2^−(deltaCt)^, where deltaCt is defined as (Ct-Experimental – Ct-Control) (*N* = 4).

### 2.6. ROS Assay

2′,7′-Dichlorodihydrofluorescein diacetate (H_2_DCF-DA) was used to identify ROS. H_2_DCF-DA becomes fluorescent after oxidation by ROS. Wild-type MEFs and *Klf4*-null MEFs incubated in NG were compared with those incubated in FM. Cells were stained with 10 mM H_2_DCF-DA in DMSO for 45 min, washed with PBS, photographed, and quantified. An Olympus IX51 fluorescent microscope was used to hand-count 400 cells for each sample, with *N* = 4.

### 2.7. Flow Cytometry

Flow cytometry was used to differentiate between dead and alive cells after metabolic stress. eBioscience Fixable Viability Dye (ThermoFisher Scientific, Waltham, MA, USA) was used to tag cells after 12 h of glucose starvation. The cells were read through the FITC channel of an AccuriTM Flow Cytometer (BD Biosciences, San Jose, CA, USA) (*N* = 2).

### 2.8. Statistical Analysis

Comparison of two groups was performed using Student’s *t*-test. GLUT1 localization analysis of multiple groups was performed using Pearson’s chi-square test. Data were presented as the mean ± standard deviation. *p*-Values below 0.05 were considered statistically significant (* *p* < 0.05, ** *p* < 0.01, *** *p* < 0.001).

## 3. Results

### 3.1. KLF4 Increases Energy Efficiency and Glycolytic Capacity

We began by assessing global metabolic differences between wild-type (WT) and *Klf4-*null MEFs. An ATP assay revealed that WT MEFs demonstrate increased ATP production as compared to *Klf4-*null MEFs, indicating greater overall energy production (Figure 1A). Next, we performed a seahorse metabolic assay to determine whether this altered energy production is a result of altered glycolysis or oxidative phosphorylation. In MEFs, the extracellular acidification rate (ECAR) was the same between WT and *Klf4-*null cells after the addition of glucose, indicating comparable rates of basal glycolysis (Figure 1B,C). Addition of oligomycin resulted in a greater increase in the ECAR in WT MEFs, indicating a greater glycolytic capacity (Figure 1B,D). Regardless of drug treatment, WT MEFs demonstrated a higher oxidative consumption rate (OCR) than *Klf4-*null MEFs, indicating greater levels of oxidative phosphorylation (Figure 1E,F). Dividing the OCR by the ECAR (OCR/ECAR) gives a relative ratio of dependence on oxidative phosphorylation as compared to glycolysis. *Klf4-*null MEFs exhibited a lower OCR/ECAR, indicating a more Warburg-like metabolism (Figure 1G).

To test the consistency of these findings across cell types, as well as to determine whether the overexpression of KLF4 alters metabolic activity, we ran a seahorse assay in colorectal RKO cells. These RKO cells have an inducible promoter that activates KLF4 by the addition of PA, with DMSO as a vehicle control. In RKO cells, the ECAR was greater in PA-treated cells as compared to DMSO-treated control cells after the addition of glucose, suggesting increased basal glycolysis (Figure 1H,I). Addition of oligomycin resulted in a greater increase in the ECAR in PA-treated RKO cells as compared to DMSO-treated RKO cells, indicating a greater glycolytic capacity (Figure 1H,J). Similarly to MEFs, RKO cells treated with PA demonstrated a greater OCR regardless of drug treatment (Figure 1K,L). DMSO-treated RKO cells demonstrated a lower OCR/ECAR as compared to PA-treated RKO cells, indicating a more Warburg-like metabolism in the cells expressing less KLF4 (Figure 1M).

### 3.2. KLF4 Increases Expression of Key Glycolytic Proteins

Following the finding that KLF4 expression correlates with enhanced glycolysis, we investigated the expression of key glycolytic enzymes at the protein level. Consistent with the seahorse assay, *Klf4-*null MEFs demonstrated decreased levels of the protein hexokinase2 (HK2), which is involved in the initial rate-determining step of glycolysis (Figure 2A,B). We then looked at the expression of pyruvate kinase M2 (PKM2), which is involved in the final rate-determining step of glycolysis to create pyruvate for oxidative phosphorylation. We found that WT MEFs express significantly more PKM2 than *Klf4-*null MEFs (Figure 2A,B). To obtain an understanding of the relative rates of lactate production and export, we looked at the expression of lactate export protein monocarboxylate transporter 4 (MCT4). Interestingly, MCT4 underwent a large decrease in *Klf4-*null MEFs as compared to wild-type MEFs (Figure 2A,B). These findings demonstrate that the overall expression of glycolytic proteins is reduced in cells lacking Klf4 as compared to wild-type cells.

Since basal glycolysis differed between PA-treated and DMSO-treated RKO cells, we expected to find a significant difference between the cell types at the protein level. Consistent with the seahorse assay, PA-treated RKO cells demonstrated increased levels of HK2 (Figure 2D,E). We also found that PA-treated RKO cells express significantly more PKM2 than DMSO-treated RKO cells (Figure 2D,E). MCT4 underwent a large increase in PA-treated RKO cells as compared to DMSO-treated RKO cells (Figure 2D,E). Overall, these findings demonstrate that addition of KLF4 increases the overall expression of glycolytic proteins.

To determine whether KLF4 regulates genes involved in metabolism expression at the transcription level, we used two-step RT-qPCR (Figure 2C,F). We found that a lack of KLF4 decreases the mRNA expression of PKM2 and MCT4 in MEFs (Figure 2C). We also observed that PA-treated RKO cells have increased mRNA expression of PKM2 and MCT4 as compared to DMSO-treated RKO cells (Figure 2F). It is important to note that the effect size is not as pronounced at the mRNA level as it is at the protein level in both MEFs and RKO cells (Figure 2B,C,E,F). These findings suggest that KLF4*′*s regulation of metabolic proteins is at least partially transcriptional, although the mechanism through which it does this is unclear.

Unexpectedly, GLUT1 protein and mRNA expression was no different between wild-type and *Klf4-*null MEFs as well as PA-treated and DMSO-treated RKO cells (Figure 2B,C,E,F). We found it interesting that GLUT1 expression is not influenced by upregulation of KLF4, as more glucose is needed to fuel the increased glycolytic activity.

### 3.3. KLF4 Facilitates Localization of GLUT1 to the Outer Membrane

To further investigate the potential mechanism through which KLF4-expressing cells may take in more glucose, we used immunofluorescence to visualize GLUT1′s cellular location. As a transmembrane protein, greater levels of GLUT1 on the extracellular membrane indicate increased glucose intake and therefore increased glycolytic capacity. Confocal imaging revealed that WT MEFs show greater levels of localization than *Klf4-*null MEFs in both the presence and the absence of glucose (Figure 3A,B). However, glucose starvation appeared to have no effect on GLUT1 localization in MEFs (Figure 3B). We also found increased GLUT1 localization in RKO cells treated with PA as compared to DMSO-treated RKO cells (Figure 3C,D). Interestingly, DMSO-treated RKO cells showed increased localization after glucose starvation (Figure 3C,D).

### 3.4. KLF4 Aids in Autophagic Cell Death in Response to Glucose Starvation

We observed that WT and *Klf4-*null MEFs had different rates of cell death. A morphology analysis demonstrated that WT MEFs decreased in confluency after glucose starvation as compared to *Klf4-*null MEFs, which saw minimal change in confluency after starvation (Figure 4A). Flow cytometry confirmed that this difference in confluency is, in part, due to increased cell death in WT MEFs in response to glucose starvation as compared to *Klf4-*null MEFs (Figure 4B). To determine whether the cell death is due to autophagy or apoptosis, we performed a Western blot for LC3 and caspase-3. Additionally, we used N-acetylcysteine (NAC) to determine whether KLF4′s regulation of cell death is altered by the level of ROS. Western blots for LC3 demonstrated increased autophagy in WT MEFs at the basal level as compared to *Klf4-*null MEFs, as well as further increased autophagy in response to glucose starvation as measured by LC3II/LC3I (Figure 4C,D). Furthermore, LC3 expression was not altered by the addition of NAC, indicating a ROS-independent mechanism (Figure 4C). Western blots for caspase-3 demonstrated that there is little to no expression of cleaved caspase-3 in WT MEFs but a greater amount of cleaved caspase-3 in *Klf4-*null MEFs (Figure 4C). Caspase-3 was consistent regardless of starvation or NAC, suggesting marginally increased levels of basal apoptosis in the absence of KLF4 (Figure 4C). It is important to note that rates of cell death do not differ significantly between cell types under normal glucose conditions, so this difference in apoptosis does not explain the increased cell death in WT MEFs after glucose starvation (Figure 4B).

### 3.5. KLF4 Reduces ROS Induced from Glucose Starvation

Since we found that KLF4 acts as a tumor suppressor with respect to inducing cell death following glucose starvation, we decided to investigate whether KLF4 has any other tumor-suppressing roles in response to glucose starvation. ROS are known to be produced by glucose starvation [29] so we aimed to determine the role of KLF4 in reducing ROS buildup. H_2_DCF-DA staining was used to visualize ROS levels in MEFs in response to glucose starvation (Figure 5A). The antioxidant NAC reduced ROS levels, serving as a positive control (Figure 5A,B). In MEFs, *Klf4-*null cells had higher ROS levels than WT cells (Figure 5B). The difference between ROS levels in WT MEFs and *Klf4-*null MEFs was accentuated during glucose starvation (Figure 5B). *Klf4-*null MEFs exhibited ROS levels 3.4 times higher than WT MEFs after starvation (Figure 5B).

## 4. Discussion

Cancer is a multifaceted disease, with many different causes that lead to uncontrolled cell growth. Beyond the typical six hallmarks of cancer—sustained proliferation, evasion of growth suppressors, resistance to cell death, angiogenesis, immortality, and invasion/metastasis—the Warburg effect is frequently observed in cancers [40]. The Warburg effect consists of increased glycolytic metabolism relative to oxidative metabolism, resulting in increased lactate production [20]. This increased lactate can be used by the cell to reduce apoptosis, increase migration, and increase resistance to cancer therapy [41,42,43]. Proteins like MCT4, a lactate export protein, are able to counteract the harmful effects of increased lactate accumulation. Since many cancer cell lines are known to suppress KLF4 expression, we investigated the metabolic differences that arise from altering the level of KLF4 in MEFs and RKO cells. Although this research is not a comprehensive analysis of KLF4′s metabolic function specific to cancer cells, we used cells lacking KLF4 to find that KLF4 has tumor-suppressing properties with respect to metabolism.

We find it important to discuss the limitations of our study prior to discussing the implications of our results. First, our study offers limited mechanistic explanations as to how KLF4 regulates metabolism, and rather serves as an overview that identifies several novel metabolic changes that depend on KLF4. Although these mechanisms are important to determine, we find them to be out of the scope of this study. However, we believe these mechanisms deserve further investigation. Second, there are limitations of this study due to the genomic and proteomic variability of cancer cells. Although cancer cells share common behaviors, the underlying mechanisms differ between cell types. Because of this, it is hard to generalize our findings to cancer cells without testing these findings in many different cancer cell lines, especially given the fact that we only made use of one colorectal cancer cell line. Our research provides a strong comparison of metabolic differences between cells lacking and expressing KLF4. Given the tendency of cancer cells to downregulate KLF4 [11,12,13,14,15], we hope that our findings contribute to understanding cancer metabolism despite using only one cancer cell line.

Our findings in cells lacking KLF4 suggest that KLF4 regulates the response to glucose in a manner that decreases the harmful by-products of increased glycolysis (Figure 6). KLF4-expressing cells are found to undergo higher levels of oxidative phosphorylation at basal levels despite undergoing the same amount of glycolysis, suggesting that more pyruvate is being created and transported to the mitochondria, where it is used in a more efficient manner (Figure 1). This is supported by our finding that KLF4 increases PKM2 expression, despite having a smaller effect on HK2 expression (Figure 2). Given the function of these proteins, it is likely that glycolysis occurs at similar rates under basal conditions, but a greater proportion of glycolysis substrates are converted into pyruvate.

Although we found that KLF4 increases mRNA levels of PKM2 and MCT4 in MEFs and RKOs, the effect size of these findings was not as large as the increase in PKM2 and MCT4 protein expression caused by KLF4 (Figure 2). It is unclear from our data whether KLF4 directly or indirectly affects transcription of these genes. For example, KLF4 could act as a transcriptional activator, upregulate a gene that acts as a transcriptional activator, or recruit other transcriptional activators to a gene complex. Since KLF4 acts as both a transcriptional activator and a repressor [9,10], KLF4 could also transcriptionally repress genes that inhibit transcription of these glycolytic genes. Additionally, our data suggest that the increase in protein expression may, in part, be due to some post-translational factor, since the increase in mRNA levels is not as pronounced as the increase in protein expression. Future studies are needed to determine the mechanism through which KLF4 mechanistically regulates these proteins beyond transcription, either through protein–protein interaction or through regulation of gene expression. KLF4 may directly regulate or stabilize glycolytic proteins in the cytosol [44,45], which makes more sense from the standpoint of having an immediate effect following metabolic stress. More canonically, KLF4 may upregulate transcription of a different gene responsible for post-translational stability of the glycolytic proteins. KLF4 has many transcriptional targets [46,47], some of which likely increase levels of glycolytic proteins post-transcriptionally.

High levels of PKM2 can be beneficial in preventing the Warburg effect, such as decreasing ROS and favoring the creation of pyruvate over lactate [48]. It is important to note that high levels of PKM2 can also be harmful if regulated to take its dimeric form [49]. Additionally, increased MCT4 expression suggests a greater rate of lactate export, leading to reduced cancer-promoting effects of lactate metabolism. Complementarily to these two mechanisms, KLF4 has been found to decrease lactate production through inhibition of LDHA [50]. Interestingly, we found no effect on GLUT1 expression at both the protein and the mRNA level (Figure 2). We decided to further investigate how cells take in greater amounts of glucose if GLUT expression does not increase.

Our data suggest that KLF4 is an important regulator of GLUT1 localization (Figure 3). Dysregulation of GLUT1 is associated with numerous cancers, suggesting that dysregulation of KLF4 can have potentially harmful effects [26]. Additionally, KLF4-mediated GLUT1 localization likely explains the increased glycolytic capacity of KLF4-expressing cells (Figure 1). Interestingly, WT MEFs and PA-treated RKO cells do not increase levels of GLUT1 localization following glucose starvation (Figure 3). This may suggest that a certain amount of KLF4 is required to maximally induce GLUT1 localization, and a further increase in KLF4 will not continue to increase localization. In DMSO-treated RKO cells, which express low levels of KLF4 [15], glucose starvation likely increases KLF4 to a level sufficient enough to induce GLUT1 localization (Figure 3). Alternatively, this discrepancy may be due to other cell line differences. Most importantly, the comparisons made between WT and *Klf4*-null MEFs as well as those between DMSO- and PA-treated RKOs suggest a strong dependency on KLF4 for adequate GLUT1 localization. Although the exact mechanism of KLF4′s regulation is unknown, a few postulations can be made based on KLF4′s relationship with vesicle regulation. First, ATG7 is a known mediator of unconventional autophagy-dependent exocytosis [51,52]. We have shown previously that KLF4 increases ATG7 expression to induce autophagy [4]. It is likely that KLF4′s increase in ATG7 expression can also induce exocytosis of GLUT1. Second, Rab proteins should be investigated as they heavily regulate vesicle transport [53,54]. Our previous microarray data showed a significant upregulation of Rab proteins in WT MEFs as compared to *Klf4*-null MEFs [55]. Overall, it is likely that KLF4 aids in GLUT1 localization through transcriptional activation of some other factor associated with the transport process. We plan on carrying out additional experiments to determine the mechanism through which KLF4 regulates GLUT1 localization. Overall, KLF4 seems to be an important mediator of fast glucose intake through GLUT1 localization, while increasing pyruvate through upregulation of PKM2 and decreasing lactate through upregulation of MCT4.

Our experiments regarding the effects of glucose starvation reveal that KLF4 acts as a tumor suppressor by increasing autophagic cell death (Figure 4). This finding is important because it may provide an explanation as to how KLF4 is downregulated in many cancers. It was previously discussed that early cancer cells typically undergo mass cell death in response to glucose starvation [29]. Cancers generally have mutator phenotypes, resulting in mass mutation in response to a stressor, leading to the creation of a new cell variation that is immune to that stressor [56]. These surviving cells are then able to replicate and form a tumor fully immune to that original stressor [30]. Dysregulation of KLF4′s cell death pathway could lead to more evasive cancers that are no longer killed following glucose starvation.

There are a few known pathways that may potentially explain a response of KLF4 to glucose starvation. The most common regulatory pathways in cancer cells in response to glucose starvation are ROS generation and AMP-activated protein kinase (AMPK) [29,57,58,59]. In normal cells, low glucose activates AMPK, which then activates p53 [60], a key transcriptional activator of Klf4 during DNA damage. AMPK has been described as multifaceted in tumor progression, resulting in differential regulation of its cellular targets [61]. This indicates that KLF4 may be upregulated by glucose starvation through AMPK, but this pathway’s regulatory effects may be altered in cancer cells. Alternatively, reactive oxygen species (ROS) may be responsible for eliciting a KLF4-mediated response to glucose starvation. Low glucose levels have also been found to lead to an increase in ROS [29]. Cancer cells are known to exhibit increased ROS [62]. ROS have been found to increase RAS protein [63]. This leads to an increase in ERK1/2, subsequently increasing KLF4 [63,64]. KLF4 has been identified as an immediate early gene in response to hydrogen peroxide (acts as a ROS) among many other stressors [65,66]. These findings suggest that there are many potential pathways that may regulate KLF4 in response to glucose starvation.

Investigating how KLF4 plays a role in starvation-mediated cell death is an important component of understanding its large-scale metabolic regulation. We found that KLF4 induces autophagic cell death in response to glucose starvation (Figure 4). Our observation of a slight increase in cleaved caspase-3 in *Klf4-*null MEFs is supported by our previous findings that KLF4 reduces apoptotic activity at basal levels [10,16]. However, it is important to note that we observed no significant difference in the rate of cell death at basal levels, possibly due to counterbalancing amounts of autophagy and apoptosis between WT and *Klf4*-null MEFs (Figure 4). Since we observed no difference in the level of cell death or KLF4 expression in response to antioxidant treatment, we concluded that KLF4′s starvation response is not likely ROS dependent (Figure 4).

We investigated ROS expression and found that ROS do, in fact, increase in both MEFs in response to starvation in correlation with the literature (Figure 5) [29]. Furthermore, we found much greater levels of ROS in cells lacking KLF4 (Figure 5). This supports our previous findings that KLF4 reduces ROS through mitophagy and GSTa4 [18]. Additionally, KLF4 may reduce ROS expression through its increase of PKM2 [48].

Beyond aiding the metabolic response to glucose starvation, KLF4 contributes to a cell death response that likely kills off cells that are unable to recover from the metabolic stress at hand. This connection between cell death and glucose starvation would likely make dysregulation of KLF4 especially problematic in cancer cells. Without KLF4, cells undergoing nutrient starvation would experience the oncogenic by-products of the Warburg effect; increased ROS, which would lead to the accumulation of more mutations; and an inability to kill off cells that are losing control of their cell cycle and/or genomic stability due to the effects of metabolic stress. It is important to note that these processes must be tested in additional cancer cell lines in order to confirm our speculations. Instead, we offer strong findings on the metabolic behaviors of cells lacking KLF4 compared to those that express KLF4.

Additional research must be done to determine whether the starvation response regulates through the AMPK pathway or another pathway. KLF4′s role in mediating exocytosis should be investigated further, especially to see whether this regulation is specific to just GLUT1, metabolic proteins, or other membrane proteins. It is also important to determine the mechanism through which KLF4 regulates metabolic genes at both a transcriptional and a translational level. Additionally, future studies remain necessary to investigate KLF4′s role in non-Warburg cancer metabolism, such as metabolic shuttling and lipid metabolism.

## 5. Conclusions

When put together, our findings demonstrate that KLF4 aims to rescue cells from low glucose levels by taking in more glucose in a controlled manner that favors oxidative phosphorylation over lactate accumulation. If glucose starvation is too severe, KLF4 will initiate the cell death response. Cancer cells that dysregulate KLF4 may exhibit a more Warburg-like response to stressors, in addition to resisting cell death and undergoing increased mutability due to ROS buildup. These factors allow for more evasion and replication dependent on the dysregulation of KLF4. The dysregulation of KLF4 may cause or amplify the tumor-promoting results of the Warburg effect and should be investigated as a potential therapeutic target.

## Figures and Tables

**Figure 1 cells-10-00830-f001:**
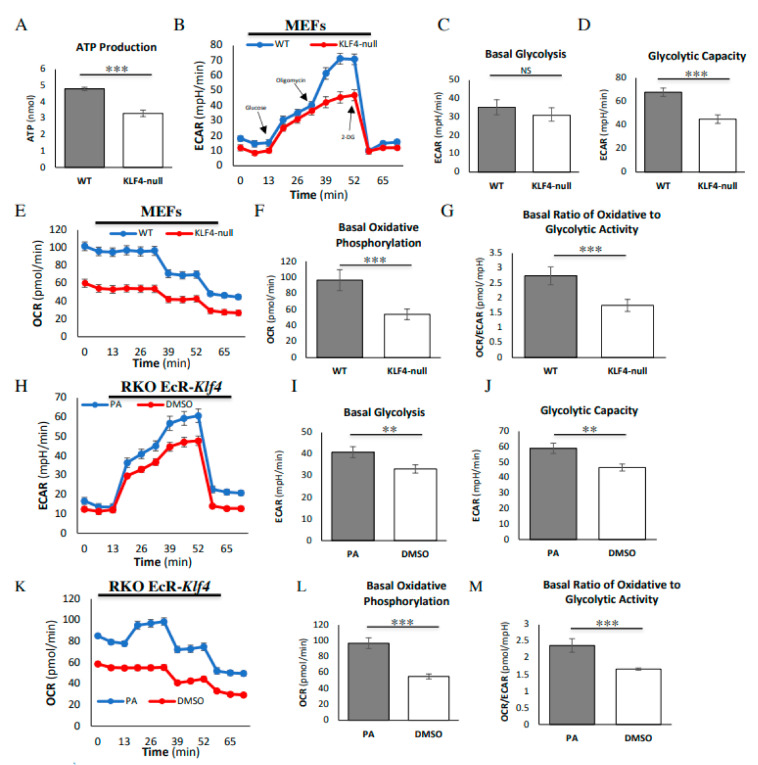
Krüppel-like factor 4 (KLF4) increases glycolytic capacity and energy efficiency. (**A**) ATP assay demonstrates increased ATP production in wild-type (WT) mouse embryonic fibroblasts (MEFs) as compared to *Klf4-*null MEFs. (**B**) Seahorse assay demonstrates an increased extracellular acidification rate (ECAR) in WT MEFs (blue) as compared to *Klf4-*null MEFs (red) in response to oligomycin. (**C**) There is no difference in basal levels of glycolysis between WT and *Klf4-*null MEFs. (**D**) WT MEFs demonstrate a greater glycolytic capacity as compared to *Klf4-*null MEFs. (**E**,**F**) Seahorse assay demonstrates an overall increase in the oxidative consumption rate (OCR) in WT MEFs (blue) as compared to *Klf4-*null MEFs (red). (**G**) WT MEFs have a greater ratio of oxidative phosphorylation to glycolysis as compared to *Klf4-*null MEFs. (**H**) Seahorse assay demonstrates an increased ECAR in KLF4-overexpressed RKO cells (blue) as compared to DMSO-treated RKO cells (red) in response to (**I**) glucose and (**J**) oligomycin. (**K**,**L**) Seahorse assay demonstrates an overall increase in the OCR in Ponasterone-A (PA)-treated RKO cells (blue) as compared to DMSO-treated RKO cells (red). (**M**) PA-treated RKO cells have a greater ratio of oxidative phosphorylation to glycolysis as compared to DMSO-treated RKO cells. Each assay was performed using three biological replicates. Error bars represent standard error of the mean. Significance was determined by Student’s *t*-test (** *p* < 0.01, *** *p* < 0.001).

**Figure 2 cells-10-00830-f002:**
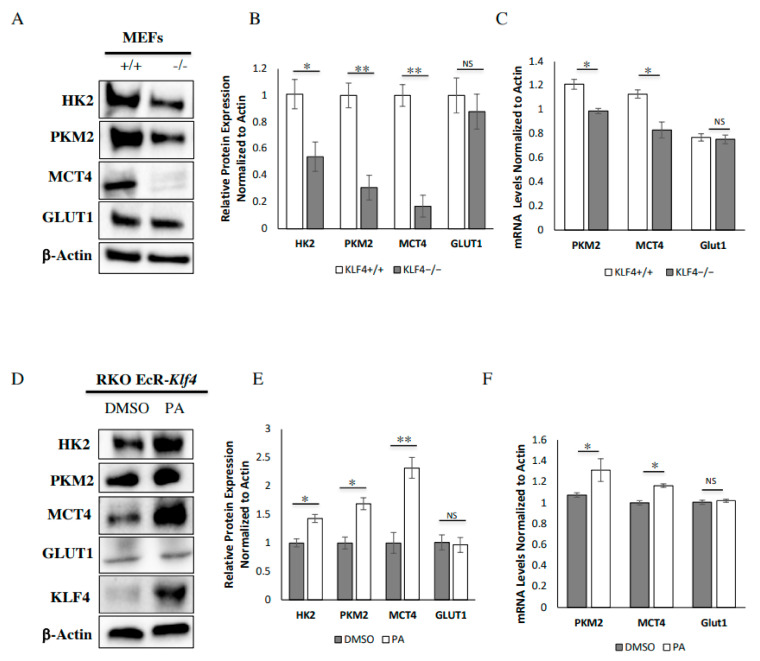
KLF4 increases expression of key glycolytic proteins. (**A**) Western blots show relative protein expression between WT and *Klf4-*null MEFs. (**B**) MEFs that do not express KLF4 have lower levels of expression of hexokinase2 (HK2), pyruvate kinase M2 (PKM2), and monocarboxylate transporter 4 (MCT4). Glucose transporter 1 (GLUT1) expression is unchanged. (**C**) RT-qPCR shows that MEFs that do not express KLF4 show lower levels of PKM2 and MCT4 mRNA. The level of GLUT1 mRNA is unchanged. (**D**) Western blots show that PA is used to overexpress KLF4 in RKO cells. (**E**) Overexpression of KLF4 increases expression of MCT4, PKM2, and HK2 in RKO cells. GLUT1 expression is unchanged. (**F**) qPCR shows that KLF4 overexpression increases levels of PKM2 and MCT4 mRNA. Levels of GLUT1 mRNA are unchanged. Each assay was performed using four biological replicates. Error bars represent standard error of the mean. Significance was determined by Student’s *t*-test (* *p* < 0.05, ** *p* < 0.01).

**Figure 3 cells-10-00830-f003:**
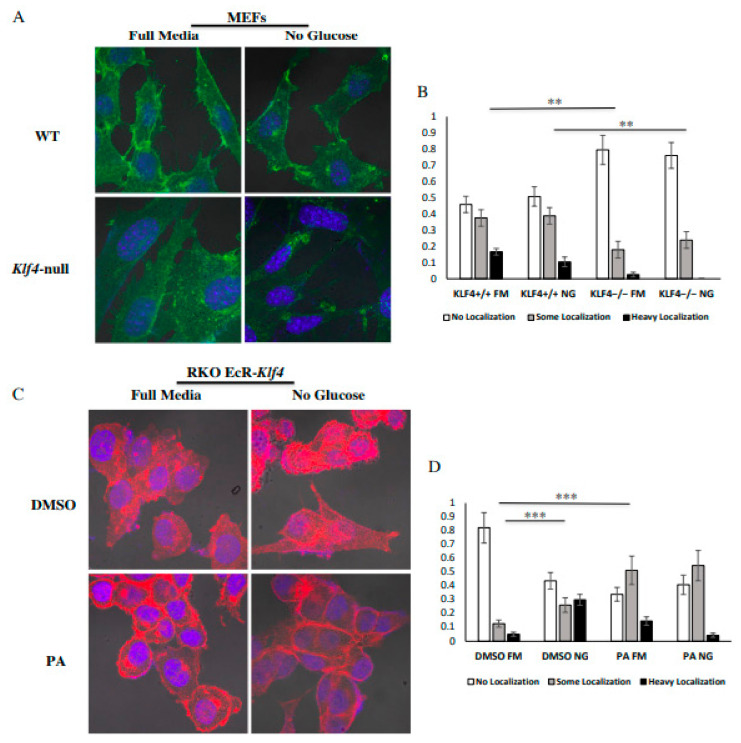
KLF4 facilitates localization of GLUT1 to the outer membrane. (**A**) Fluorescent imaging for GLUT1 (green) in MEFs. DAPI (blue) was used to stain cell nuclei. (**B**) Wild-type MEFs demonstrate greater levels of basal GLUT1 membrane localization than *Klf4-*null MEFs. Furthermore, glucose starvation has no effect on GLUT localization in either cell type. (**C**) Fluorescent imaging for GLUT1 (red) in RKO cells. DAPI (blue) was used to stain cell nuclei. (**D**) PA-treated RKO cells demonstrate greater levels of GLUT1 membrane localization as compared to DMSO-treated RKO cells. Furthermore, glucose starvation increases membrane localization of GLUT1 in DMSO-treated RKO cells, while there is no difference in PA-treated RKO cells. For MEFs, 45–55 cells were counted for each treatment over three biological replicates (at least 15 cells counted in each replicate). For RKO cells, 95–105 cells were counted for each treatment over three biological replicates (at least 30 cells counted in each replicate). Error bars represent standard error of the mean. Significance was determined by Pearson’s chi-square test (** *p* < 0.01, *** *p* < 0.001).

**Figure 4 cells-10-00830-f004:**
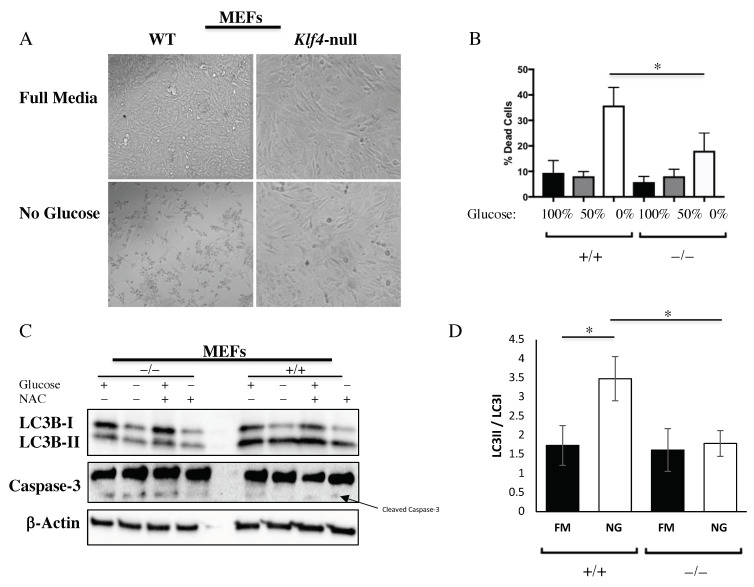
KLF4 aids autophagic cell death in response to glucose starvation in an N-acetylcysteine (NAC)-independent manner. (**A**) MEF morphology. (**B**) WT MEFs die at a greater rate as compared to *Klf4-*null MEFs in response to glucose starvation. (**C**,**D**) KLF4 induces autophagic cell death in MEFs in response to starvation, independent of reactive oxygen species (ROS). Each assay was performed using four biological replicates. Error bars represent standard error of the mean. Significance was determined by Student’s *t*-test (* *p* < 0.05).

**Figure 5 cells-10-00830-f005:**
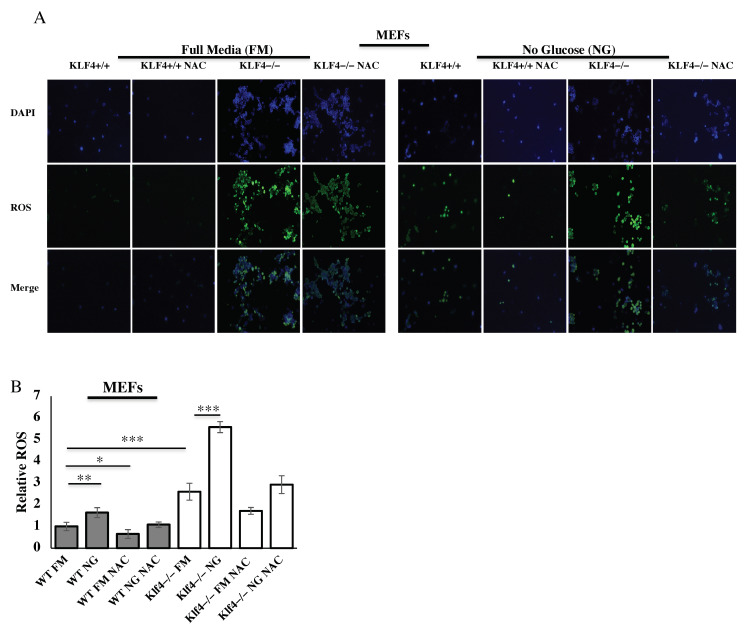
KLF4 decreases ROS from glucose starvation. (**A**) Fluorescent imaging for ROS used 2′,7′-dichlorodihydrofluorescein diacetate (H_2_DCF-DA) (green) in MEFs. DAPI (blue) was used to stain cell nuclei. (**B**) KLF4 reduces ROS, especially during starvation where ROS are increased significantly. Each treatment condition was repeated with three biological replicates. Error bars represent standard error of the mean. Significance was determined using Student’s *t*-test (* *p* < 0.05, ** *p* < 0.01, *** *p* < 0.001).

**Figure 6 cells-10-00830-f006:**
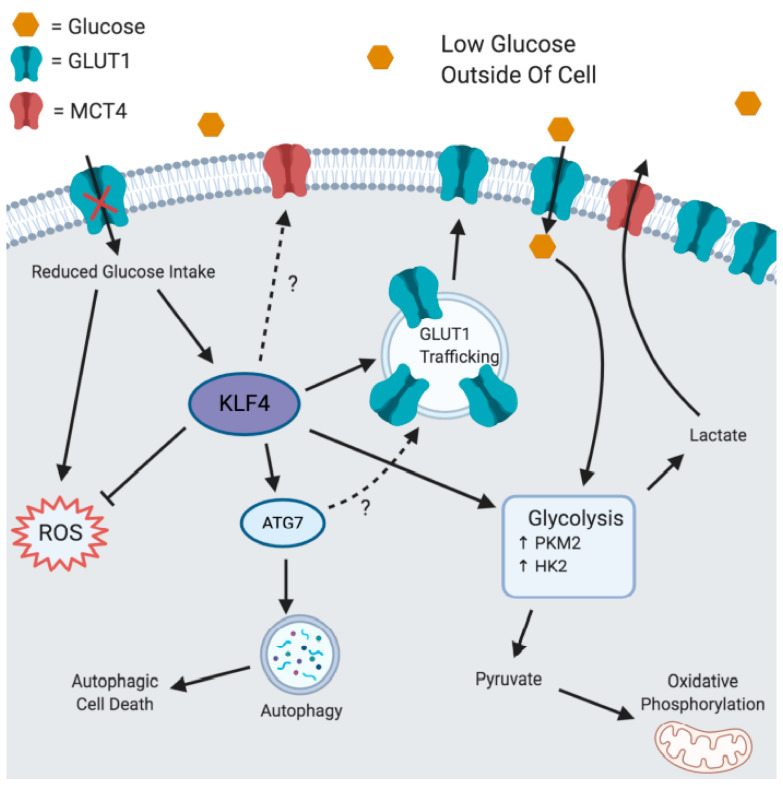
KLF4′s role in responding to glucose starvation. Reduced glucose intake increases ROS as well as KLF4. KLF4 protects the cell by inhibiting ROS and inducing autophagic cell death. Furthermore, KLF4 responds to reduced glucose intake by facilitating GLUT1 localization to the cellular membrane. Additionally, KLF4 increases PKM2 to favor oxidative phosphorylation and increases MCT4 to export lactate out the cell. Created using BioRender.

## Data Availability

The datasets used and/or analyzed during the current study are available from the corresponding author on reasonable request.

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
