# Peer review of "KLF4 Regulates Metabolic Homeostasis in Response to Stress"

_cells, 2021, doi:10.3390/cells10040830_

Round 1
Reviewer 1 Report
Blum et al described the effect of KLF4 in regulation of metabolic homeostasis in response to stress. They showed that KLF4 can rescue cells from low glucose levels by taking in more glucose in a controlled manner that favors oxidative phosphorylation over lactate accumulation. KLF4 can induce the cell death response in terms of glucose starvation. Also, they concluded that the dysregulation of KLF4 may be the cause of tumor promotion results of the Warburg effect.
In general the manuscript has an acceptable flow, however, the details and quality of figures need to be improved
1)Figure 1 (C,D,F,G,I,J,L,M), the author should verify the followings
a) What does the error bar mean? mean+/- SD or SEM ,...etc
b) How many independent experiments from which these results are derived?
2) Figure 2:
a) The authors need to verify why the translational not transcriptional levels are affected by KLF4?
b) Figure 2B,2C, 2E, 2F: The same comments as in 1: What does the error bar mean? and How many independent experiments from which these results are derived?
c) Why the authors did not do HEK2 transcript level by qPCR? (Figure 2C)
d) In figures 2C and 2F: no error bars are presented in some column
3) Figure 3B, 3D: There is no error bars in the column, and how the authors calculate the statistics on these figures
4) Figure 4: the same comments for figure 1 and figure 2 for figure 4b, 4c (right)
Reviewer 2 Report
In this manuscript Blum et al. investigate the role of KLF4 in the context of glucose starvation.
As a model system the authors use mouse embryonic fibroblasts that are either KLF4 +/+ or KLF4 -/- and RKO, a colon cancer derived cell line that harbors an inducible KLF4 construct. By a combination of western blot, qPCR and immunofluorescence they show that 1. KLF4 expression correlates with enhanced glycolysis as shown by upregulation of key glycolytic proteins as well as localization of GLUT1 to the outer membrane and 2. KLF4 induces autophagic cell death in response to glucose starvation.This data clearly demonstrates that that KLF4 regulates the response to glucose starvation in the cells tested.
Overall the manuscript is well written and the results provide an interesting potential direction for future studies in the field.
If the following comments are properly addressed I will recommend this manuscript for publication in “Cells”:
- In the manuscript the authors generalize their conclusion on the role of KLF4 to “cancer cells”. However, their results are only based on one cancer cell line tested (RKO). Therefore there is not enough evidence for extending their conclusions. I recommend to either use additional cancer cell lines to provide a stronger support to some of the key experimental results or to tone down their conclusions when referring to “cancer cells” in general.
- The qPCR results in Figure 2 show that there is no effect on mRNA expression of key glycolytic enzymes in the presence/absence of KLF4. However Figure 1 shows a clear change in their protein expression level pattern that is dependent on KLF4 . Although the authors effectively conclude that KLF4’s regulation of metabolic proteins is post-translational, the molecular mechanism is not clear. Blum et al. should try to further investigate this point or provide some kind of speculative explanation in the Discussion section. One possibility is that KLF4, although not directly regulating the transcription of the tested glycolytic enzymes could however influence the transcription of other factors that might then modulate the glycolytic enzymes protein stability. An easy experiment to check this hypothesis would be to repeat the experiment showed in Figure 2A in the presence of an RNA Polymerase II inhibitor.
- Figure 3 shows that KLF4 expression has an effect on the localization of GLUT1 to the extracellular membrane. The mechanism for this effect has not been addressed. If the transcriptional activity of KLF4 is not required for GLUT1 localization then the authors could check if there is a protein-protein interaction between KLF4 and GLUT1 by a simple co-immunoprecipitation assay. Alternatively, a mass-spec experiment could elucidate if GLUT1 interacts with differential protein complexes in the absence or presence of KLF4 (by comparing WT MEF to KLF4 null MEFs and by comparing RKO cells to RKO+PA).
- Finally, it is not clear why in Figure 3 there is no GLUT1 localization to the extracellular membrane upon glucose starvation. The authors should provide some kind of explanation.
Round 2
Reviewer 1 Report
I read carefully the replies to my comments, and I checked the revised manuscript. I do not have any further concerns
Author Response
We were very pleased to see that our manuscript received a conditional acceptance. We very much agree with the concern to describe the limitations of our study. We added a discussion of the limitations as the second paragraph of our discussion (page 13, lines 345-358). Our changes can be seen using track changes in our newly submitted manuscript. Additionally, we have pasted it here:
We find it important to discuss the limitations of our study prior to discussing the implications of our results. First, our study offers limited mechanistic explanations as to how KLF4 regulates metabolism, but rather serves as an overview which identifies several novel metabolic changes that depend on KLF4. Although these mechanisms are important to determine, we find it to be out of the scope of this study. However, we believe these mechanisms deserve further investigation. Secondly, there are limitations to this study due to the genomic and proteomic variability of cancer cells. Although cancer cells share common behaviors, the underlying mechanisms differ between cell types. Because of this, it is hard to generalize our findings to cancer cells without testing these findings in many different cancer cell lines, especially given the fact that we only made use of one colorectal cancer cell line. Our research provides a strong comparison of metabolic differences between cells lacking and expressing KLF4. Given the tendency for cancer cells to downregulate KLF4 [11,12,13,14,15], we hope that our findings contribute to understanding cancer metabolism despite using only one cancer cell line.
We believe that there are two limitations that are important to address, especially in response to reviewer 2’s initial concerns. First, we discussed the limitation in mechanistic explanations. We explain that despite discovering novel regulatory effects of KLF4, we do not look into how this is done (page 13 lines 346-350). Secondly, we discussed the limitations of our study with respect to applying our results to cancer (page 13 lines 350-358). We explain that the increased variability in cancer cells means that we would need to utilize many cancer cell lines in order to make definitive claims. Instead, we used cells lacking KLF4 to predict cancer metabolism given the findings that KLF4 is downregulated in many cancer cells.
We find it important to discuss our limitations to make it clear that we understand the extent to which our work can be applied to cancer. Furthermore, describing our limitations makes a clearer framework for what future research should be done. We thank you very much for your recommendations which have strengthened our paper.
We have very much enjoyed the publication process that Cells has offered, being very quick to review and respond to our concerns. We look forward to future collaboration with this journal.
Sincerely,
Engda Hagos